# Impact of the COVID-19 Pandemic on Pediatric Emergency Medicine: A Systematic Review

**DOI:** 10.3390/medicina58081112

**Published:** 2022-08-17

**Authors:** Chien-Wei Cheng, Yan-Bo Huang, Hsiao-Yun Chao, Chip-Jin Ng, Shou-Yen Chen

**Affiliations:** 1Department of Emergency Medicine, Chang Gung Memorial Hospital, Keelung and Chang Gung University College of Medicine, Taoyuan City 333, Taiwan; willcheng77@gmail.com; 2Department of Emergency Medicine, Chang Gung Memorial Hospital and Chang Gung University, Linkou, Taoyuan City 333, Taiwan; yanhusuo79619@gmail.com (Y.-B.H.); b101092070@tmu.edu.tw (H.-Y.C.); ngowl@ms3.hinet.net (C.-J.N.); 3Graduate Institute of Clinical Medical Sciences, Division of Medical Education, College of Medicine, Chang Gung University, Taoyuan City 333, Taiwan

**Keywords:** pediatric emergency department, COVID-19, pediatric patient volume, pediatric trauma patient

## Abstract

(1) *Background and Objectives:* The COVID-19 pandemic has considerably affected clinical systems, especially the emergency department (ED). A decreased number of pediatric patients and changes in disease patterns at the ED have been noted in recent research. This study investigates the real effect of the pandemic on the pediatric ED comprehensively by performing a systematic review of relevant published articles. (2) *Materials and Methods:* A systematic review was conducted based on a predesigned protocol. We searched PubMed and EMBASE databases for relevant articles published until 30 November 2021. Two independent reviewers extracted data by using a customized form, and any conflicts were resolved through discussion with another independent reviewer. The aggregated data were summarized and analyzed. (3) *Results:* A total of 25 articles discussing the impact of COVID-19 on pediatric emergencies were included after full-text evaluation. Geographic distribution analysis indicated that the majority of studies from the European continent were conducted in Italy (32%, 8/25), whereas the majority of the studies from North America were conducted in the United States (24%, 6/25). The majority of the studies included a study period of less than 6 months and mostly focused on the first half of 2020. All of the articles revealed a decline in the number of pediatric patients in the ED (100%, 25/25), and most articles mentioned a decline in infectious disease cases (56%, 14/25) and trauma cases (52%, 13/25). (4) *Conclusions:* The COVID-19 pandemic resulted in a decline in the number of pediatric patients in the ED, especially in the low-acuity patient group. Medical behavior changes, anti-epidemic policies, increased telemedicine use, and family financial hardship were possible factors. A decline in common pediatric infectious diseases and pediatric trauma cases was noted. Researchers should focus on potential child abuse and mental health problems during the pandemic.

## 1. Introduction

The COVID-19 outbreak, which was first identified in the Wuhan city of China in December 2019, rapidly spread to the rest of the world within the next few months. Because of its severity, the World Health Organization declared the COVID-19 outbreak as a pandemic on 11 March 2020 [1]. Thus far, COVID-19 has caused more than 6 million deaths and infected more than 545 million individuals globally [2].

With the progression of the pandemic, the patient volume of the emergency department (ED) has continuously decreased, especially regarding the number of pediatric patients, in many countries [3,4,5,6,7,8,9]. Moreover, this situation was noted in some countries with a low severity of the pandemic [10]. This decrease may be partly attributable to parents’ fear of being infected or anti-epidemic policies, including wearing a mask, social distancing, and school closure, which have reduced the spread of diseases that typically infect children [9,11,12]. All of these factors have contributed to the continued decrease in the number of pediatric patients in the ED.

The timing of COVID-19 outbreaks and the severity of the pandemic vary among regions, and countries may have had different policies for the pandemic. This study investigates the real effect of the COVID-19 pandemic on the pediatric ED by performing a systematic review of relevant published articles and a further qualitative analysis of outcomes. Our study findings may help health officials or hospital operators worldwide to develop appropriate policies and help clinical physicians understand changes in the number of pediatric patients in the ED and the incidence of common infectious diseases during the pandemic.

## 2. Materials and Methods

### 2.1. Systematic Review Protocol

A systematic review was performed using a predesigned protocol in accordance with the Preferred Reporting Items for Systematic Reviews and Meta-Analyses (PRISMA) statement [13].

### 2.2. Search Strategy

We searched the PubMed and EMBASE databases for relevant articles from the beginning of 1 November 2019 to 30 November 2021. The search strategy was based on the following algorithm: ((“emergency medical services” [MeSH Terms] OR emergency health service [Text Word]) AND (“pediatric” [MeSH Terms] OR pediatric [Text Word])) AND (“COVID-19” [MeSH Terms] OR coronavirus disease 2019 [Text Word]) for PubMed, and emergency health service AND pediatric AND coronavirus AND disease AND 2019 for EMBASE. 

### 2.3. Inclusion and Exclusion Criteria

We included all original studies discussing the pediatric patients in the ED during the COVID-19 pandemic period. Only articles published in English were included. Review articles, case reports, non-original articles, and letters to editors were excluded from this study.

### 2.4. Study Selection and Data Extraction

Figure 1 presents the PRISMA diagram of the study selection and review process. After identifying relevant studies from PubMed and EMBASE, we manually removed duplicate articles. Two independent reviewers (CWC and YBH) examined the titles and abstracts of all the identified articles. On the basis of inclusion and exclusion criteria, articles were further excluded. We evaluated the full text of articles that had a title but no abstract. Only articles that were eliminated by both the reviewers were removed from the list. Subsequently, the two reviewers thoroughly read the full text and extracted the data of the articles by using a customized form (Appendix A) during the evaluation stage. During the full-text evaluation, an Excel sheet was created to collate essential information on the articles. Information on the title, first author, article type, year of publication, journal of publication, and journal category was extracted and listed in the Excel sheet. Any disagreement between the two reviewers was resolved through discussion with a third reviewer (SYC). We used Newcastle-Ottawa Scale to assess the quality of the articles (Appendix A). The included articles were qualitatively analyzed and summarized narratively.

### 2.5. Statistical Analysis

Descriptive statistics of aggregated data are presented as numbers and proportions. The included articles were analyzed according to the country/geographic location, study content, and study period. Descriptive analysis was performed using Microsoft Excel (2016, Microsoft Corporation, Seattle, WA, USA).

## 3. Results

We identified 229 articles from PubMed and 203 articles from EMBASE (Figure 1). After the removal of duplicate articles and the evaluation of titles and abstracts, 113 articles were included. A total of 25 articles remained after the full-text evaluation. Table 1 lists the detailed information of these 25 articles. They are listed in alphabetical order by the name of the first author.

Figure 2 presents a world map depicting the country of origin for each article. Most of the studies were conducted in Italy in the European continent (32%, 8/25). Most of the studies were conducted in the United States for North America (24%, 6/25). Two studies (8%) collected data from multiple countries. No study conducted in South America or Africa focused on COVID-19 and its effect on the number of pediatric patients in the ED. 

Figure 3 presents the time frame of the included studies. The majority of the studies had a study period of less than 6 months and mostly focused on the first half of 2020. Ten studies had a study period of more than 6 months (40%), and only four studies (16%) had their study period extended to the whole year. 

Table 2 presents the study content of the included studies. All of the studies indicated declines in the number of pediatric patients in the ED (100%, 25/25). Some of the studies revealed the decline in number in infectious disease departments (56%, 14/25) and trauma departments (52%, 13/25). A total of 12 studies (48%) revealed an increase in the ward admission rate for pediatric patients during the pandemic. The intensive care unit (ICU) admission rate was reported in ten studies, and no change in the ICU admission rate was observed in most of the studies. Notably, some studies pointed to an increase in child abuse cases (16%, 4/25) [7,22,27,29]. Table 3 lists factors contributing to a decreased volume of pediatric patients in the ED.

## 4. Discussion

In this review, we noted a decreasing overall trend in the number of pediatric patients in the ED during the COVID-19 pandemic. However, in terms of geographical distribution, the articles included in our studies were not evenly conducted across the globe. Most of the studies were conducted in the United States of North America because the United States is a powerhouse of scientific research. In Europe, the majority of the studies were conducted in Italy. This was probably because Italy was the first country in Europe to have a major COVID-19 outbreak and was the hardest hit region in Europe [6,27]. Excluding North America and West Europe, one study was conducted in the Middle East and four studies were conducted in Asia. No studies were performed in Central/South America or Africa. Less motivation to publish articles with similar conclusions or a lack of research power or funding are possible reasons for this. Although the trend of a decrease in the number of pediatric patients in the ED was noted globally, a conclusion still requires further support by data from these areas. 

Although the study period varied among different articles, most of the studies included a study period of a few months, particularly Italian studies. Most of their articles were published early in the pandemic, coinciding with the Italian national lockdown from 11 March 2020 to 4 May 2020, and these studies investigated the effect of the national lockdown. Studies conducted in other countries had similar conditions and typically ended before August 2020. The results of these studies in the early phase of the pandemic only reflect the short-term impact of the pandemic and anti-epidemic policies. To investigate the long-term effect of the COVID-19 pandemic on pediatric ED patients, the results of studies conducted for over a year are warranted. Furthermore, studies with longer periods may be necessary to understand the persistent effect of the pandemic, which could change the epidemiology of pediatric patients.

All the studies demonstrated a decrease in the number of pediatric patients in the ED during the COVID-19 pandemic. A possible reason for this finding could be anti-epidemic policies, such as the national lockdown. The results of Italian studies demonstrated a decline in the number of pediatric patients in the ED before, during, and after the lockdown and revealed the sharpest decline during the lockdown period [6,24]. The number of pediatric patients in the ED increased after the lockdown period but did not return to the pre-COVID 19 period [6]. Another possible reason for the decrease in the number of pediatric patients in the ED was the fear of COVID-19 exposure in the hospital setting. Parents may avoid seeking hospital-based services due to the fear of COVID-19 infection. A web-based survey performed in Chicago determined that approximately 25% of caregivers were hesitant to bring their children for ED care [33]. The majority of the studies reported a prominent decline in the number of pediatric patients in the ED and an increased hospitalization rate during the pandemic [6,8,12,16]. Delays in sending children with acute illnesses to the ED resulted in a more advanced stage of their illness, and a study recommended clinical physicians to remain alert regarding delayed treatment due to the fear of COVID-19 [34]. Government officers and pediatricians should publicize the greater risk of delayed medical treatment over contracting COVID-19 [17]. The increased use of telemedicine during the pandemic was another factor that contributed to a decline in the number of pediatric patients in the ED [25,27,35]. During the initial outbreak of the epidemic in Italy, patients with fever and respiratory symptoms were suggested to call their general practitioners and follow instructions based on their clinical condition instead of visiting the ED directly. Pediatric patients and their parents contacted their primary care physicians to address acute medical problems through virtual means during the epidemic. One study reported a 154% increase in telehealth visits during the pandemic [36]. The regional ambulance service helped transport patients to the hospital based on the judgment of physicians by telemedicine [14]. This finding may explain the more commonly observed arrival to the ED by an ambulance in some studies. Moreover, economic changes caused by the pandemic may be responsible for the decline in the number of pediatric patients in the ED. Financial hardship and unemployment may lead many patients to avoid seeking medical care due to concerns regarding the inability to afford expenses [25]. A disproportionate decline in pediatric ED visits for potentially vulnerable children characterized by race and ethnicity and insurance status may be attributed to the financial factor [16].

Another crucial trend was the reduction in common pediatric infectious diseases. A decline in infectious diseases involving the respiratory and gastrointestinal tracts has been reported in previous studies [7,12]. School closure and widespread social distancing reduced opportunities for the transmission of common infectious diseases. Using a face mask and hand washing reduced the route of droplet transmission and fecal–oral transmission. Moreover, the prevalence of some non-infectious respiratory illnesses, such as asthma exacerbation, decreased significantly [25]. National or regional lockdown during the COVID-19 pandemic resulted in a marked reduction in air pollution because of decreased traffic, industrial activity, and office heating [37]. These factors contributed to decreased air pollution which, in turn, reduced the exacerbation of asthma and the occurrence of respiratory infections in children [38].

A decline in pediatric trauma cases during the pandemic was reported in half of the included articles [6]. Most trauma occurred in the outdoor setting due to limited outdoor activities, such as motor vehicle travel. Moreover, playground closure and the cancellation of sporting events during the epidemic reduced the risk of injury [25]. However, a study conducted in Canada reported that trauma-related ED visits in children aged <10 years increased during the pandemic [18]. This may be related to accidents in the home environment, such as falls and burns, because limited school attendance prolonged their time staying at home. The under-supervision of young children resulting from some caregivers concurrently working from home contributed to increased injuries [39]. 

Child abuse was another crucial problem in terms of pediatric trauma cases, although no study reported an increase in the number of visits due to child abuse during the pandemic. Home isolation during lockdown may induce the events of child abuse because isolation increases the risk of domestic violence and neglect [40]. Parental factors, including job loss, anxiety, burnout and depression, can serve as contributory factors [41,42]. The prevention of home injury and domestic accidents should be considered in the pandemic.

A higher proportion of children with mental health disorders among pediatric ED visits during the pandemic was noted in one study [22]. Imposed quarantines may exacerbate existing mental health problems due to social isolation [43]. Moreover, mental health services for adolescents with lower household incomes may be disrupted due to school closures [44]. A New York study reported a doubling number of ED visits for self-injury, suicidal ideation, or suicide attempts [31]. In terms of mental health and the physical development of children and adolescents, an appropriate social life should be maintained as much as possible while considering infection control measures and anti-epidemic policies [45]. 

The decline in pediatric patient visits has severely affected the economics of sustaining a pediatric ED. The cancellation of non-urgent and elective care visits and rising costs associated with disrupted medical supply chains has left many US healthcare systems in financial crisis [46]. How long the epidemic will last and whether pediatric ED volume will return remains unclear. Thus, the adjustment of the medical workforce and the reallocation of resources based on changes in pediatric ED visits are essential. 

### Limitations

This study has several limitations. First, we only included studies published in English. Thus, some essential articles written in different languages could have been ignored. Second, we did not add other terms such as “paediatric”, or “child/children” in our search. This could lead to incomplete search results and missing articles. Third, the articles included in our research were mostly published in the United States and Italy. Inadequate data from other areas could render the conclusion insufficiently comprehensive. Fourth, we only searched two online databases in our study, so some relevant articles may not be included in our analysis. Fifth, we only included articles conducted within 2 years of the COVID-19 pandemic, so some long-term effects may not have been found and reported. Finally, the measures for infection control and medical habits were diverse in different countries, and this could induce inconsistency in the results of the studies.

## 5. Conclusions

The COVID-19 pandemic has considerably reduced the number of pediatric patients in the ED globally, especially in low-acuity patient groups. Changes in medical behavior due to the fear of COVID-19; anti-epidemic policies, such as national lockdown; increased telemedicine use; and family financial hardship are possible contributory factors. A decline in common pediatric infectious diseases has been noted and is attributed to anti-epidemic policies, including mask wearing, social distancing, school closure, and limited activities. The number of pediatric trauma cases has reduced, except domestic accidents. Potential child abuse and mental health problems during the pandemic should be given attention while maintaining infection control measures and physical distancing.

## Figures and Tables

**Figure 1 medicina-58-01112-f001:**
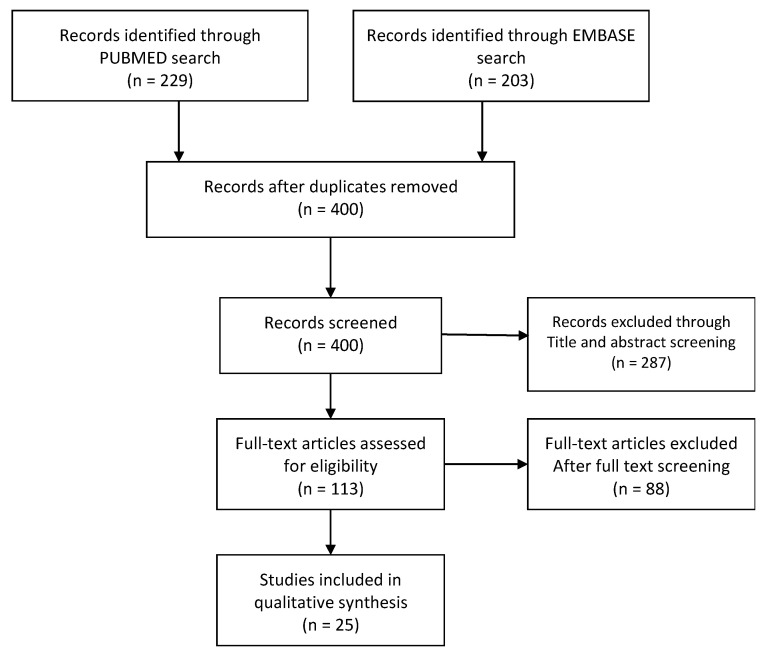
PRISMA flow diagram of the study selection process.

**Figure 2 medicina-58-01112-f002:**
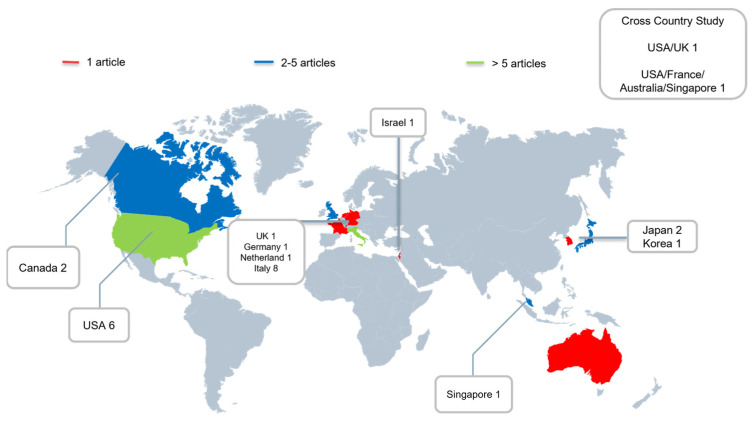
Geographic distribution of the included articles.

**Figure 3 medicina-58-01112-f003:**
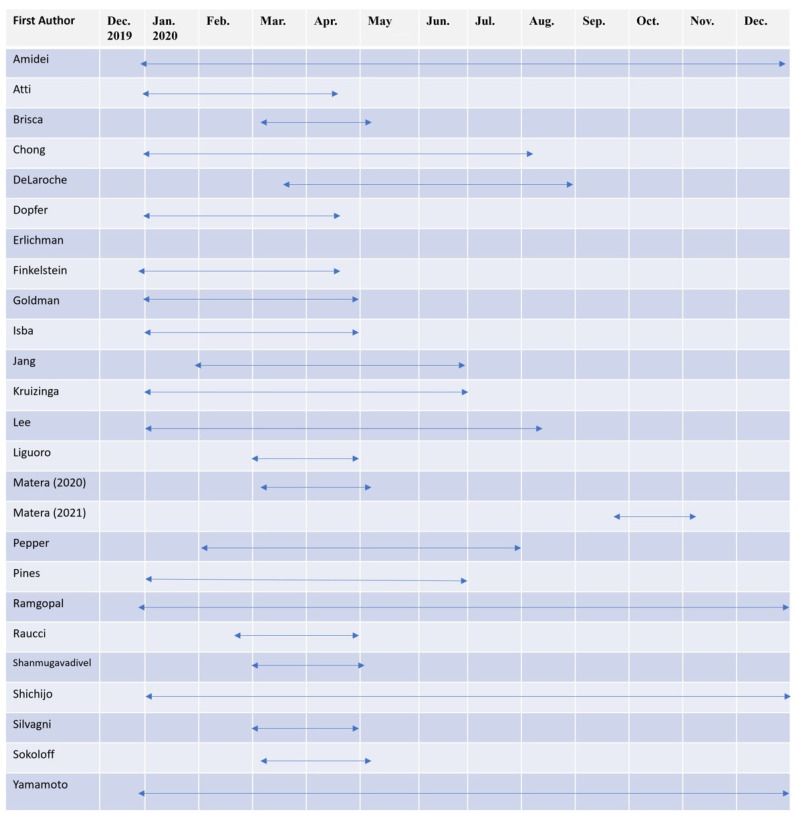
Timeline of study periods of the included articles.

**Table 1 medicina-58-01112-t001:** Key information of the included original articles.

	First Author (Year)	Title	Primary Outcome	Geographic Location	Study Setting
1	Amidei (2021) [6]	Pediatric emergency department visits during the COVID-19 pandemic: a large retrospective population-based study	ED presensataion reduced; Hospitalization increased	Italy	Multicenter
2	Atti (2020) [14]	Facing SARS-CoV-2 Pandemic at a COVID-19 Regional Children’s Hospital in Italy	ED visits and urgent hospitalization reduction	Italy	Single hospital
3	Brisca (2021) [15]	The impact of COVID-19 lockdown on children with medical complexity in pediatric emergency department	ED visits declined; Hospitalization increased	Italy	Single hospital
4	Chong (2020) [7]	Impact of COVID-19 on pediatric emergencies and hospitalizations in Singapore	Mean ED attendance and infectious diseases declined; Trauma-related diagnosis declined; Child abuse cases increased	Singapore	Single hospital
5	DeLaroche (2021) [16]	Pediatric Emergency Department Visits at US Children’s Hospitals During the COVID-19 Pandemic	ED visits and infectious diseases declined; Child abuse cases decreased	USA	Multicenter
6	Dopfer (2020) [17]	COVID-19 related reduction in pediatric emergency healthcare utilization—A concerning trend	ED visits decreased but admission rate increased; Decline in infectious disease	Germany	Single hospital
7	Erlichman (2021) [8]	The ongoing indirect effect of the COVID-19 pandemic on a pediatric emergency department	ED visit declined; Hospitalization rate increased	Isreal	Single hospital
8	Finkelstein (2021) [18]	Effect of the COVID-19 Pandemic on Patient Volumes, Acuity, and Outcomes in Pediatric Emergency Departments: A Nationwide Study	Reducion in ED visits and mental health; Increase in ward, ICU admission and trauma cases	Canada	Multicenter
9	Goldman (2020) [12]	Paediatric patients seen in 18 emergency departments during the COVID-19 pandemic	ED visits declined; Admission rate increased	Canada	Multicenter
10	Isba (2020) [19]	COVID-19: Transatlantic Declines in Pediatric Emergency Admissions	ED volume declined; Higher odds of admission	UK/USA	Multicenter
11	Jang (2021) [20]	Pediatric Emergency Department Utilization and Coronavirus Disease in Daegu, Korea	ED volume declined; Higher hospitalization rate	Korea	Multicenter
12	Kruizinga (2021) [21]	The impact of lockdown on pediatric ED visits and hospital admissions during the COVID19 pandemic: a multicenter analysis and review of the literature	ED visits, admission and infectious dieases declined	Netheralands	Single hospital
13	Lee (2021) [9]	Paediatric ED utilisation in the early phase of the COVID-19 pandemic	ED volume, hospitalization and ICU admission declined	Australia/France/Singapore/USA	Multicenter
14	Liguoro (2021) [22]	The impact of COVID-19 on a tertiary care pediatric emergency department	ED visitis declined; Child abuse cases increased	Italy	Single hospital
15	Matera (2020) [23]	SARS-CoV-2 Pandemic Impact on Pediatric Emergency Rooms: A Multicenter Study	ED admission decreased as with infectious diseases	Italy	Multicenter
16	Matera (2021) [24]	Effects of relaxed lockdown on pediatric er visits during SARS-CoV-2 pandemic in Italy	Reduction in ED visits and infectious diseases	Italy	Multicenter
17	Pepper (2021) [25]	Analysis of pediatric emergency department patient volume trends during the COVID-19 pandemic	ED visits, behavioral health and fractures declined	USA	Single hospital
18	Pines (2021) [11]	Characterizing pediatric emergency department visits during the COVID-19 pandemic	ED visits and infectious dieases declined	USA	Multicenter
19	Ramgopal (2021) [26]	Forecast modeling to identify changes in pediatric emergency department utilization during the COVID-19 pandemic	ED encounter lowered; Trauma and infectious diseases below prediction	USA	Multicenter
20	Raucci (2021) [27]	Impact of the COVID-19 pandemic on the Emergency Department of a tertiary children’s hospital	Reduction in ED visits; Doubling of relative frequency of hospitalization; Child abuse cases increased	Italy	Single hospital
21	Shanmugavadivel (2021) [28]	Changing patterns of emergency paediatric presentations during the first wave of COVID-19: Learning for the second wave from a UK tertiary emergency department	ED attendance declined	UK	Single hospital
22	Shichijo (2021) [29]	Patient attendance at a pediatric emergency referral hospital in an area with low COVID-19 incidence	ED outpatient declined; Ward and infectious dieases declined; Psychological interventions increased; Child abuse cases increased	Japan	Single hospital
23	Silvagni (2021) [30]	Neonatal and pediatric emergency room visits in a tertiary center during the COVID-19 pandemic in Italy	ED visits, infectious dieases and accident visits declined; Hospital admission reduced while ICU remained same	Italy	Single hospital
24	Sokoloff (2021) [31]	Pediatric emergency department utilization during the COVID-19 pandemic in New York City	Reduction in ED visits; Admission rate increased; Suicide-related visits and self-harm increased by 100% Child abuse cases declined	USA	Single hospital
25	Yamamoto (2021) [32]	Pediatric emergency healthcare utilization during the COVID-19 pandemic in Tokyo	ED utilization declined	Japan	Single hospital

COVID-19: coronavirus disease 2019; ED: emergency department; USA: United States of America; UK: United Kingdom; SARS-CoV-2: severe acute respiratory syndrome–related coronavirus 2.

**Table 2 medicina-58-01112-t002:** Summarized effects of COVID-19 on pediatric emergency department.

Total N = 25	Increase	Decrease	No Change	Not Specified
Pediatric ED volume	0 (0%)	25 (100%)	-	0 (0%)
Infectious disease cases	0 (0%)	14 (56%)	-	11 (44%)
Ward admission rate	12 (48%)	7 (28%)	1 (4%)	5 (20%)
ICU admission rate	2 (8%)	2 (8%)	6 (24%)	15 (60%)
Trauma cases	1 (4%)	13 (52%)	-	11 (44%)
Mental health cases	2 (8%)	4 (16%)	1(4%)	17 (68%)
Child abuse case	4 (16%)	2 (8%)	-	19 (76%)

COVID-19: coronavirus disease 2019; ICU: intensive care unit.

**Table 3 medicina-58-01112-t003:** Contributory factors to reduced pediatric emergency department volume.

Medical behavior change: fear of contracting COVID-19 infection (72%, 18/25) [6,7,9,12,15,17,19,20,21,22,23,24,25,27,29,30,31,32]
Anti-epidemic policies: national lockdown (68%, 17/25) [6,7,9,12,14,15,16,17,18,21,22,23,24,25,26,28,32]
Decreased common pediatric infectious disease transmission: wearing masks, school closure, hand washing, social distancing (72%, 18/25) [6,7,8,11,12,14,15,18,19,20,22,23,24,25,27,29,30,32]
Increased telemedicine use rather than direct ED visit (24%, 6/25) [8,11,14,25,27,28]
Financial hardship: unemployment (4%, 1/25) [25]

## Data Availability

The dataset supporting the conclusions of this article is included within the article.

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
