# Peer review of "Impact of the COVID-19 Pandemic on Pediatric Emergency Medicine: A Systematic Review"

_medicina, 2022, doi:10.3390/medicina58081112_

Round 1

Reviewer 1 Report

This manuscript describes a systematic review analyzing the impact of the COVID-19 pandemic on the volume of pediatric patients admitted to a pediatric emergency room during the pandemic. Using the PRISMA statement, the authors searched PubMed and EMBASE and finally included 25 articles. This is an interesting article with important findings. I believe that some clarifications would help enhance the quality of this manuscript even further.

Major comments

1. According to the inclusion criteria, the Authors "included all original studies discussing the volume of pediatric patients in the ED during the COVID-19 pandemic period. This a is an important topic, but this objective is not clearly stated. The Title and the end of the Introduction section should be adapted to make this objective much clearer.

2. While "pediatric" was included as [Text word] in the search, "paediatric" was not. This should be acknowledged as a clear limitation and may have prevented the identification of relevant articles. Indeed, adding "paediatric [Text Word]" to the second term of the search (“pediatric” [MeSH Terms] OR pediatric [Text Word] OR paediatric [Text Word]) rather than just (“pediatric” [MeSH Terms] OR pediatric [Text Word]) led to the identification of 20 more articles (14%, 20/141) in a quick searched performed on the day of this review.

3. In line with the previous comment, I obtained 141 article (July 31st) by running the exact string provided by the Authors, who however claimed that they "identified 229 articles from PubMed". I therefore have trouble understanding this difference. Could the Authors please provide an explanation (were other strings used?)?

4. Data extraction using a customized form is only mentioned in the abstract. It should also be explained within the core text, and a sample form should be provided as supplementary material

5. Table 3 is most interesting, but would be even more impactful if the articles reporting these contributory factors were listed (this would also help provide the "N" of articles reporting each factor.

6. In the discussion, the authors report that "Child abuse was another crucial problem in terms of pediatric trauma cases". However, there is no data regarding this critical issue in the "Results" section. Data regarding child abuse should appear in the results section before being discussed.

Minor comments

7. Some references should be reformatted (e.g., refs 1&2 are currently referenced as "Organization, W.H." while "World Health Organization" would be more appropriate)

8. The PRISMA diagram is much too small and is currently hard to decipher. A higher-quality graphic should be provided. To this end, the Authors could use the templates provided here: https://prisma-statement.org//PRISMAStatement/FlowDiagram

9. When giving percentages, I would advise the use of (x%, n/N). This would give a clearer idea of the actual findings.

10. Table 1: please change "Frist" to "First"

11. Table 1: The journal and SCI category do not provide much information. Conversely, the primary outcome and the geographical setting (already well shown thanks to the excellent Figure 2) would be more informative, and I believe that these variables should be included rather than the former ones.

Author Response

Major comments

  1. According to the inclusion criteria, the Authors "included all original studies discussing the volume of pediatric patients in the ED during the COVID-19 pandemic period. This a is an important topic, but this objective is not clearly stated. The Title and the end of the Introduction section should be adapted to make this objective much clearer.

Ans: Thank you for your comment. Our statement about the inclusion criteria may lead some misunderstandings. Although patient volume is an important topic, our study wanted to include the original articles discussing all aspects of pediatric patients in the ED during the pandemic rather than “patient volume” only. We have corrected the statement of inclusion criteria. Please see Line 76. We apologize for the mistake. Thank you!

  1. While "pediatric" was included as [Text word] in the search, "paediatric" was not. This should be acknowledged as a clear limitation and may have prevented the identification of relevant articles. Indeed, adding "paediatric [Text Word]" to the second term of the search (“pediatric” [MeSH Terms] OR pediatric [Text Word] OR paediatric [Text Word]) rather than just (“pediatric” [MeSH Terms] OR pediatric [Text Word]) led to the identification of 20 more articles (14%, 20/141) in a quick searched performed on the day of this review.

Ans: Thank you for your valuable suggestion. The term “ paediatric” should be added into our search strategy to make the result more complete. However, we may not be able to redo the search and review because of the time problem. We hope you could understand. We also have added this limitation in our acknowledgement according to your suggestion. Please see Line 236-237.

  1. In line with the previous comment, I obtained 141 article (July 31st) by running the exact string provided by the Authors, who however claimed that they "identified 229 articles from PubMed". I therefore have trouble understanding this difference. Could the Authors please provide an explanation (were other strings used?)?

Ans: Thank you for your comment. We tried to search again by using our string and we got the result of 311 articles (2019/11/1-2022/07/31). We are not sure why there is a difference. We have attached our search result. Please see the picture.

  1. Data extraction using a customized form is only mentioned in the abstract. It should also be explained within the core text, and a sample form should be provided as supplementary material

Ans: Thank you for your suggestion. We have added the description in the main text and provided the sample of customized form (Supplementary file 1). Please see Line 87-88.

  1. Table 3 is most interesting, but would be even more impactful if the articles reporting these contributory factors were listed (this would also help provide the "N" of articles reporting each factor.

Ans: Thank you for your suggestion. We have revised the table by adding citations, percentage/number according to your suggestion. Please see Table 3.

  1. In the discussion, the authors report that "Child abuse was another crucial problem in terms of pediatric trauma cases". However, there is no data regarding this critical issue in the "Results" section. Data regarding child abuse should appear in the results section before being discussed.

Ans: Thank you for your comment. We have added the data regarding “Child abuse” into the Result section and Table 2. Please see Line 126 and Table 2.

Minor comments

  1. Some references should be reformatted (e.g., refs 1&2 are currently referenced as "Organization, W.H." while "World Health Organization" would be more appropriate)

Ans: Thank you for your comment. We have revised the references and corrected the errors. Thank you!

  1. The PRISMA diagram is much too small and is currently hard to decipher. A higher-quality graphic should be provided. To this end, the Authors could use the templates provided here: https://prisma-statement.org//PRISMAStatement/FlowDiagram

Ans: Thank you for your comment. We have revised the Figure 1. Thank you!

  1. When giving percentages, I would advise the use of (x%, n/N). This would give a clearer idea of the actual findings.

Ans: Thank you for your suggestion. We have revised the presentation according to your suggestion. Thank you!

  1. Table 1: please change "Frist" to "First"

Ans: Thank you for your correction. We have corrected the typo. Thank you!

  1. Table 1: The journal and SCI category do not provide much information. Conversely, the primary outcome and the geographical setting (already well shown thanks to the excellent Figure 2) would be more informative, and I believe that these variables should be included rather than the former ones.

Ans: Thank you for your suggestion. We have revised our table according to your suggestion. Please see Table 1.

Reviewer 2 Report

Introduction

-          Well-written with concise information

Methodology

-          Is the study protocol registered? E.g.: PROSPERO

-          Searching only two online databases render the study to a huge limitation. I would strongly suggest to at least search for 5 databases.

-          Did the authors consider other relevant keywords, such as ‘child’ or ‘children’?

-           It is July 2022 now, perhaps the authors should extend the search to at least January 2022.

-          The exclusion criteria should be more precise. Is review papers accepted or rejected?

-          Statistical analysis was performed using Microsoft Excel” What kind of statistical analysis did the authors perform?

-          The authors might want to perform risk of bias analysis for each included study

Results

-          I would highly suggest the authors to put citation in table 1.

-          Table 3 lists factors contributing to a decreased volume of pediatric patients in the ED” Where did those factors originated from? Which studies?

-          I do not fully understand the significance of Table 2. Are the authors trying to mention whether paedatric patients increase or decrease in numbers during the pandemic? And which studies mentioned increase? As well as which studies mentioned decrease? A transparency of the data information is highly appreciated.

Discussion

-          I agreed with most of the discussion points.

Author Response

Introduction

-  Well-written with concise information

Ans: Thank you for your comment.

Methodology

-  Is the study protocol registered? E.g.: PROSPERO

Ans: Thank you for the question. We did not register the study protocol. We will register the protocol if we have similar study in the future.

-  Searching only two online databases render the study to a huge limitation. I would strongly suggest to at least search for 5 databases

Ans: Thank you for your suggestion. Although we would like to revise the search result according to the suggestion, it may be difficult to redo the search and review due to inadequate time. We hope you could understand our difficulty. We have added this into our limitations. Please see Line 239-240. Thank you for your valuable suggestion again!

-  Did the authors consider other relevant keywords, such as ‘child’ or ‘children’?

Ans: Thank you for your comment. We did not use the keywords “ child/children” in our search. This may induce incomplete search result. We have added this into our limitations. Please see Line 236-237.

-  It is July 2022 now, perhaps the authors should extend the search to at least January 2022.

Ans: Thank you for your suggestion. We have done the search until 2021/11 because we wanted to do a search after a complete 2-year period since the outbreak of COVID-19. Actually, we started to perform the review and analysis since 2021/12 and it took much time. There would always be a time gap between search and article completion. We hope you could understand. Thank you!

-  The exclusion criteria should be more precise. Is review papers accepted or

rejected?

Ans: Thank you for your suggestion. The review articles were excluded in our study. We have revised the manuscript to make it more clear. Please see Line 77-78.

-  “Statistical analysis was performed using Microsoft Excel” What kind of statistical

analysis did the authors perform?

Ans: Thank you for your comment. Only descriptive analysis was done by using Excel. We have revised the text. Please see Line 99.

-  The authors might want to perform risk of bias analysis for each included study

Ans: Thank you for your comment. As our known, bias analysis is usually performed to assess randomized controlled studies in systematic review and meta-analysis. Most of the included articles in our study were observational studies, and our study was more like a scoping review, which may not need bias analysis. However, we still tried our best and perform the bias analysis by using Newcastle-Ottawa Scale. Please see Supplementary file 2. Thank you!

Results

-  I would highly suggest the authors to put citation in table 1.

Ans: Thank you for your suggestion. We have added the citations into the table. Please see Table 1.

-  “Table 3 lists factors contributing to a decreased volume of pediatric patients in the ED” Where did those factors originated from? Which studies?

Ans: Thank you for your comment. We have added the citations of each factor. Please see Table 3.

-   I do not fully understand the significance of Table 2. Are the authors trying to

mention whether paedatric patients increase or decrease in numbers during the pandemic? And which studies mentioned increase? As well as which studies mentioned decrease? A transparency of the data information is highly appreciated.

Ans: Thank you for your comment. Our study aimed to know the impact of COVID-19 comprehensively. Although some effects showed consistence in different countries, some could be different because of varied epidemic, cultures, national policies. We displayed table 2 to explain the consistency of the influence. For example, most articles showed decreased patient volume and infectious diseases identically. However, the results of mental health, child abuse, and admission rate were diverse. We want to let the readers know which effect was consistent globally and which outcome could be different individually. For the transparency of the data information, we revised our table 1 and showed the primary outcome and major findings. We hope this revision could improve the transparency. Thank you!

Discussion

-   I agreed with most of the discussion points.

Ans: Thank you for your comment.

Round 2

Reviewer 1 Report

Dear Authors,

Thank you for providing us with this revised version of your manuscript. I only have one more minor comment:

Minor comment 1: at the end of the sentence "Notably, some studies points to an increase in child abuse 126 cases (16%, 4/25)", adding the references would be appreciated since it would help their retrieval (and this outcome is not reported in Table 1).

Author Response

Minor comment 1: at the end of the sentence "Notably, some studies points to an increase in child abuse 126 cases (16%, 4/25)", adding the references would be appreciated since it would help their retrieval (and this outcome is not reported in Table 1).

ANS: Thank you for your suggestion. We have added the references and this outcome in Table 1. Please see Line 129 and Table 1.

Reviewer 2 Report

Since the study only evaluate the included studies qualitatively, I highly recommend to remove 'and analysis' from the title. In fact, the PRISMA chart also shown studies included for qualitative analysis. 

Otherwise, all comments have been addressed adequately.

Author Response

Since the study only evaluate the included studies qualitatively, I highly recommend to remove 'and analysis' from the title. In fact, the PRISMA chart also shown studies included for qualitative analysis.

Otherwise, all comments have been addressed adequately.

ANS: Thank you! We have revised the title and removed 'and analysis'.